# Emotion Regulation Is Associated with Anxiety, Depression and Stress in Adults with Cerebral Palsy

**DOI:** 10.3390/jcm12072527

**Published:** 2023-03-28

**Authors:** Ingrid Honan, Emma Waight, Joan Bratel, Fiona Given, Nadia Badawi, Sarah McIntyre, Hayley Smithers-Sheedy

**Affiliations:** 1Cerebral Palsy Alliance Research Institute, Specialty of Child & Adolescent Health, The University of Sydney, Camperdown, NSW 2050, Australia; 2Australian Centre for Health, Independence, Economic Participation and Value Enhanced Care for Adolescents and Young Adults with Cerebral Palsy (CP-Achieve), Melbourne, VIC 3052, Australia; 3Grace Centre for Newborn Care, The Children’s Hospital at Westmead, Specialty of Child & Adolescent Health, The University of Sydney, Westmead, NSW 2145, Australia

**Keywords:** cerebral palsy, disability, emotion regulation, depression, anxiety

## Abstract

Emotion regulation difficulties are associated with many neurological conditions and negatively impact daily function. Yet little is known about emotion regulation in adults with cerebral palsy (CP). Our aim was to investigate emotion regulation in adults with CP and its relationship with condition-related and/or socio-demographic factors. In a cross-sectional study of adults with CP, participants completed a survey containing the Difficulties in Emotion Regulation Scale (DERS), Depression Anxiety and Stress Scale-21 (DASS-21), and socio-demographic and condition-related questions. Descriptive statistics, chi-squared and Mann–Whitney tests were performed. Of the 42 adults with CP (x31.5 years, SD13.5) that were tested, 38 had within normal limits DERS total scores; however, a significantly higher proportion of participants experienced elevated scores (i.e., more difficulties with emotion regulation) than would be expected in the general population across five of the six DERs subdomains. Moderate–extremely severe depression and anxiety symptoms were reported by 33% and 60% of participants, respectively. The DERS total scores for participants with elevated depression, anxiety, and stress scores were significantly higher than the DERS totals score for those without elevated depression, anxiety, and stress scores. DERS and DASS-21 scores did not differ significantly by condition-related nor socio-demographic characteristics. In conclusion, emotion regulation difficulties were associated with elevated symptoms of depression and anxiety, which were overrepresented in the adults with CP participating in this study.

## 1. Introduction

Cerebral palsy (CP) is an umbrella term for a group of non-progressive motor disorders resulting from damage to or maldevelopment of the immature brain [1]. These disorders are permanent but not unchanging and are frequently accompanied by other conditions, including emotional and behavioural difficulties [1]. Preschool children with CP have been shown to experience more emotional/behavioural difficulties than their peers [2]. Similarly, a large European study of primary school aged children with CP found almost one third had emotion difficulties and 40% had significant social impairments as a result of their emotional or behavioural problems [3]. Other studies of school aged children with CP from Australia and Canada have also identified higher rates of emotional, behavioural, and social difficulties compared with typically developing children [4,5]. A recent study comparing associations between emotion knowledge and emotion regulation abilities of 36 children with CP with 45 typically developing children demonstrated that children with CP exhibited poorer emotion knowledge and poorer emotion regulation skills than their typically developing peers, and that there was an association between emotion regulation abilities and emotion knowledge [6]. However, less is known about how emotional difficulties manifest and impact people with CP in adulthood.

Emotion regulation (ER) refers to how we ‘influence which emotions we have, when we have them, and how we experience and express them’ [7]. It involves a series of processes, including: a situation, cognitively attending both to the situation and our emotions, appraising our emotions in the situation, and developing a response. Being able to intrinsically experience and regulate our emotions provides us with a feeling of control and allows us to respond to the demands of daily life in a socially acceptable manner [8]. This is useful in education, employment, and in building and maintaining relationships across a lifespan [9,10,11].

ER difficulties are associated with a range of conditions. More than half of all psychological diagnostic categories (e.g., anxiety and mood disorders) and all personality disorders include ER difficulties as part of the symptomatology required to meet diagnostic criteria [7,12]. ER difficulties are also associated with executive functions and attention deficit hyperactive disorder (ADHD). Given people with CP are more likely to have co-occurring ADHD than the general population, and people with ADHD are at increased risk of experiencing psychological symptoms in adulthood such as anxiety and depression, examining the occurrence of ER difficulties in adults with CP is warranted.

The neural underpinning of ER is also important to examine when considering the likelihood of associations between ER difficulties and CP in adulthood. A model proposed by Etkin et al. (2015) describes how predictions, prediction errors, and valuations contribute towards two forms of ER: model-free regulation and model-based regulation [13]. However, the way in which the neural systems and associated brain regions involved in ER are affected in many conditions is still unclear. It is acknowledged that the dorsal anterior cingulate, insula, amygdala, and periaqueductal grey are all associated with emotional reactivity. Similarly, ER evidence from fields such as Tourette Syndrome, depression, anxiety, traumatic brain injury, and Parkinson’s disease suggest that components of the basal ganglia, the amygdala, and/or the connections between these regions play a key role in emotion regulation [14,15]. However, little is known about CP and ER. Some forms of CP, particularly dyskinetic CP, are caused by damage to the basal ganglia [16]. Moreover, brain pathology for people with CP occurs in early development [17], likely impacting plasticity and experiential learning pathways. A recent study examining neural circuitry in adults with CP demonstrated associations between level of functional connectivity in brain regions associated with ER and social skills, and participant well-being scores [18]. As regions of the brain known to control ER have been associated with CP, it is reasonable to hypothesise that adults with CP may experience an increased incidence of ER difficulties than expected in the general population.

Research is required to better understand ER and the relationship between ER and clinical and socio-demographic factors in adults with CP. This is necessary to determine whether there is a need for services and screening to support adults with CP and to assist in managing these challenges. In this study, we aimed to (1) investigate self-reported ER difficulties in adults with CP, and (2) to determine whether ER was associated with other demographic and condition-related factors, including symptoms of mood difficulties (depression, anxiety, and stress), ambulatory status, communication level, living arrangements, employment status, etc. It was hypothesised that (1) adults with CP are more likely to experience ER difficulties than the general population, and (2) that ER difficulties are associated with mood and condition-related factors and socio-demographic factors such as ambulatory status and employment status, respectively.

## 2. Materials and Methods

### 2.1. Study Design

This was a cross-sectional study design. The research question was initiated by co-author FG who has a lived experience of CP. Key stages of study design and interpretation of results were informed by lived experience and current best research practice. Methodology and data are reported according to the CHERRIES (Checklist for Reporting Results of Internet E-Surveys) statement [19].

### 2.2. Ethics

Ethical approval was granted by the Cerebral Palsy Alliance Human Research Ethics Committee (3 November 2018), Victorian Cerebral Palsy Register Governance, and Queensland Cerebral Palsy Register Steering Committee, with data collected throughout 2019. All participants were provided a participant information sheet outlining study protocol and informed consent was obtained.

### 2.3. Participants, Procedures and Measures

The study was advertised in the NSW/ACT CP Register newsletter and via a study flyer (Appendix B Figure A1). To be eligible, participants were required to have a confirmed diagnosis of CP, be aged ≥ 18 years, and have no/mild intellectual impairment as recorded in the CP register. No incentives were offered for participation in the study.

Participants completed a voluntary 15-min paper or online survey. The survey included questions related to socio-demographic and clinical condition, including gender, age, residential postcode, employment status, living arrangements, primary support person, Gross Motor Function Classification System-Expanded and Revised (GMFCS-E&R) [20] level, predominant CP motor type, Communication Function Classification System (CFCS) [21] level, use of Augmentative and Alternative Communication (AAC), receipt of supports/services, and history of any psychological disorder/s, e.g., anxiety/depression. Descriptors were provided alongside classification systems such as the GMFCS and CFCS to allow participants to self-select their level if they did not already know it. Moreover, questions around communication and AAC were asked using branching logic, and participant responses were checked for consistency. See Appendix A for a full copy of the survey.

Other items included the Depression Anxiety and Stress Scale short-form (DASS-21) [22] and the Difficulties of Emotion Regulation Scale-36 (DERS) [23]. The DASS-21 is a validated self-report measure of psychological distress widely used in research and clinical practice. Using a four-point Likert scale 0–3 (never, sometimes, often, almost always) individuals rate their responses to 21 items. Seven items contributed to each of the sub-scales: depression, anxiety, and stress. Cut-off scores have been developed for each sub-scale to indicate mild, moderate, severe, and extremely severe scores. Cut-off scores indicate symptoms relative to the general population, rather than level of severity of a diagnosable disorder, i.e., ‘mild’ indicates mildly elevated symptoms of depression compared to the general population, not that the individual meets diagnostic criteria for depression in the mild range. The DASS-21 was adapted from the DASS-42 and has demonstrated adequate construct validity and high reliability (0.88 depression, 0.82 anxiety, and 0.90 stress) [22].

DERS is a validated 36-item self-report measure of emotion regulation. Using a five-point Likert scale 1–5 (almost never, sometimes, about half of the time, most of the time, almost always) participants rate responses. Responses were tallied to obtain a total score and six sub-scale scores: non-acceptance of emotional responses (nonaccept), difficulties engaging in goal directed behaviour (goals), impulse control difficulties (impulse), lack of emotional awareness (awareness), limited access to ER strategies (strategies), and lack of emotional clarity (clarity) [23]. Higher scores indicate greater difficulties with ER. Total and sub-scale scores can be converted into age and gender adjusted T-scores, with a mean of 50 and a standard deviation (SD) of 10 [24]. Validity and reliability of DERS is well established, with good internal consistency (Cronbach’s α ranging from 0.80 to 0.89) and test–retest reliability for total DERS scores being 0.88 [23].

Online open survey data were collected using REDCap electronic data capture tool, hosted by The University of Sydney. A response was required for all items on the DASS-21 and DERS and participants were able to progress through the survey and review their previous responses at their own pace. The survey was completed by five individuals prior to study commencement to review functionality and acceptability. A copy of the survey is available in Appendix A. In line with ethical requirements, participants who endorsed items suggesting that they were a safety risk to themselves or others were contacted by a registered psychologist and offered support and/or links to services.

### 2.4. Statistical Analyses

Employment was collapsed into two categories based on participant responses: ‘Employed’ = full-time, part-time, casual, volunteer, student; ‘Unemployed’ = unemployed, retired. Similarly, living situation was collapsed into four categories: ‘Alone’; ‘Share house’ = share house or supported accommodation; ‘With spouse’ = with or without children; ‘Other family’ = living with parents, living with other children alone or living with other family. GMFCS was collapsed into two categories: ‘Ambulant’ = GMFCS level I or II; ‘Supported mobility’ = GMFCS levels III, IV and V. DASS-21 scores were categorised into normal, mild, moderate, severe, and extremely severe groups [22]. Scores were then further collapsed into two groups: “Low” = normal and mild scores and “Elevated” = moderate, severe, extremely severe, based on normative data. “Elevated” scores therefore indicate that an individual endorses feelings of depression equal to or more often/severe than 88 percent of the normative sample, feelings of anxiety equal to or more often/severe than 92 percent of the normative sample, and feelings of stress equal to or more often/severe than 89 percent of the normative sample [22]. These cut-offs were selected as they are considered clinically significant. DERS-36 total scores and sub-scale scores were converted to age and gender adjusted T-scores [24]. DERS T-scores were categorised into two groups; “WNL” (within normal limits) = those who scored ≤ 1.5 SD above the mean (a T-score of ≤65), and “Elevated” = those who scored more than 1.5 SD above the mean (a T-score of >65).

Data were analysed using SPSS (Version 24). Descriptive and frequency statistics were run for all variables. Means and proportions were calculated for DASS-21 and DERS sub-scales, and chi-squared tests were performed on DERS scores to compare whether the current sample proportions differed from expected population proportions, based on normative data. Mann–Whitney tests were conducted to examine the relationship between Total DERS scores and DASS-21 sub-scales, with significance set to *p* = 0.05. Mean total DERS T-scores by DASS-21 sub-scale category were calculated and plotted onto a bar graph. Where sample sizes allowed (*n* ≥ 5), Mann–Whitney tests were performed to examine relationships between clinical and socio-demographic factors and DASS-21 and DERS scores.

## 3. Results

After eliminating any duplicate responses, there were 48 respondents who participated in the survey; two were ineligible, two declined consent, and two were incomplete. There were *n* = 42 participants with complete data which were included in the analyses. Table 1 outlines the participant characteristics.

The mean age of participants was 31.5 years (SD 13.48) and 43 percent were males. The majority of participants had spastic hemiplegia and were ambulant (Table 1). Most participants reported that they were effective communicators with no intellectual impairment. When examining socio-demographic factors, *n* = 24/42 (58%) of participants were living with other family members, with the majority of this group living with their parents. Seventy percent of participants were engaged in some form of employment (including volunteer work or study); however, only 22.5 percent of participants were working full-time. Just over 40 percent of participants had received a formal diagnosis of a psychological disorder.

Examination of self-reported ER using the DERS found that only four participants reported experiencing total elevated DERS scores. Due to small sample size in the elevated group, it was not possible to examine statistical significance. However, when examining sub-scale scores, a higher proportion of participants experienced elevated scores across all sub-scale areas, except lack of emotional clarity, than would be expected in the general population (Table 2).

When examining results from the DASS-21 (Table 2) the data for symptoms of depression indicated that 67% percent of participants reported symptoms in the ‘normal–mild’ range with 33% above this range (moderate–extremely severe). In terms of anxiety, 40% reported symptoms of anxiety in the ‘normal–mild’ range with the remaining 60% with reported symptoms in the ‘moderate–extremely severe’ range. The majority (88%) of participants reported symptoms of stress in the ‘normal–mild’ range.

There was a significant positive relationship between depression, anxiety, and stress, and total DERS scores. DERS total scores for participants with elevated depression, anxiety, and stress scores was significantly higher than the DERS totals score for those without elevated depression, anxiety, and stress scores (Table 3, Figure 1).

When examining DERS total T-scores, depression, anxiety, and stress scores by gender and employment status, there were no significant differences across groups (Table 4). Similarly, when examining factors relating to motor condition, there was no significant difference in scores across groups for gross motor function, communication level, or whether or not participants were receiving routine services. Those who had a history of a diagnosed psychological disorder obtained significantly higher DERS total T-scores, anxiety, and stress scores than those who had no history of a diagnosed psychological disorder. There was no difference between groups for depression scores.

## 4. Discussion

To date, there has been little research investigating ER in adults with CP. To begin to explore these factors we conducted a cross-sectional survey of adults with CP with no/mild intellectual impairment. Self-reported ER scores varied widely, with 90% of participant total scores falling within normal limits. However, when examining ER sub-scale scores, a higher proportion of participants scored in the elevated range across all sub-scales, except emotional clarity, than would be expected in the general population. Worryingly, moderate-extremely severe symptoms of depression and anxiety were common; however, most participants’ stress scores fell within normal limits. A clear positive relationship was observed between mood and ER scores, whereby people with elevated depression, anxiety, and stress scores experienced more ER difficulties than those without elevated depression, anxiety, and stress scores. Apart from having a diagnosed psychological disorder, no other clinical or socioeconomic factors were associated with the DERS or DASS-21 scores in this sample.

A statistically significant proportion of the total group had elevated sub-scale scores for nonacceptance of emotional responses (*n* = 10, 24%), difficulties engaging in goal-directed behaviour (*n* = 6, 14%), impulse control difficulties (*n* = 6, 14%), lack of emotional awareness (*n* = 6, 14%), and limited access to ER strategies (*n* = 7, 17%). Lower acceptance of emotions, not having strategies, or using less effective ER strategies have been associated with higher depressive symptoms [25,26]. This relationship between ER and mood disorders has been observed in other diagnostic groups, including individuals with traumatic brain injury and eating disorders, [27,28] and suggests that use of effective ER strategies (e.g., mindfulness practices and meditation) may have a role to play in the management of mood disorders [29].

It is understood that adults with CP experience higher rates of mood disorders than the general population [30,31,32]. Here we found that one third of participants had depression scores falling in the moderate to extremely severe range and more than half had anxiety scores in the moderate to extremely severe range. Whilst data are not diagnostic in nature and therefore not directly comparable, this is considerably higher than what might be expected from population estimates of 10% percent of Australians with depression and 13% percent with anxiety disorder [33]. Interestingly, adults with CP in this study were no more likely to experience stress than the general population. Whilst stress of caregivers of people with CP has been well researched, evidence examining stress in adults with CP is scarce.

The multi-factorial mechanisms responsible for high rates of depression and anxiety in CP are not well understood. In some instances, the brain injury or maldevelopment responsible for an individual’s CP may also predispose them to mood disorders and emotion dysregulation [30]. Brain injuries or maldevelopments common in CP can impact the frontal cortex, cerebellum, and limbic system, all of which have important roles in how we regulate emotions, mood, and behaviour [30,34]. Moreover, given that the underlying brain injury or maldevelopment responsible for a person’s CP occurs in or before infancy, and ER development occurs across childhood, adolescence, and young adulthood, early injury may have downstream developmental effects. Beyond a biological predisposition, there are a number of other environmental factors that may contribute to mood disorders and associated emotion dysregulation in CP such as pain, social isolation, and marginalisation.

Pain and mood disorders such as depression often co-occur in the general population, with the relationship between pain and mood being bidirectional [35,36]. Chronic pain is reported in both adults and children with CP at a higher rate than the general population, with studies across the world reporting that 33–84% of participants with CP experience pain [30,37,38,39]. Whilst there is little research describing the contribution of pain to depression in CP, management of pain is essential to support both a good quality of life and to promote mental well-being.

Social isolation and loneliness are also risk factors for mood disorders [40,41]. Whilst not investigated in this study, there is research to suggest that adults with CP experience more social isolation and loneliness than adults without disability [42]. It is hypothesised that this relates to barriers faced in regards to social integration, forming relationships, finding employment, communication, accessing transport for activities outside the home, and having limited accommodation options/choice, all of which can contribute to social isolation [41,42,43]. In addition to improving quality of life, strategies to support social connectedness may reduce the risk of mood disorders and associated emotion dysregulation in CP.

### Strengths and Limitations

To the best of our knowledge, this is the first study to investigate the relationship between ER, mood, and socio-demographic and condition related factors in adults with CP. The research question for this study was originally initiated by author FG who has a lived experience of CP. We found that adults with CP in this study had increased experiences of emotional dysregulation than the general population, had elevated symptoms of depression and anxiety, and that increasing symptoms of depression, anxiety, and stress were associated with poorer emotion regulation. In terms of limitations, this was a preliminary study which aimed to examine ER across adults with CP, and no matched comparison group was included in the design. We did not specifically target any CP motor subgroups for recruitment. The small number of respondents meant it was not statistically viable to investigate ER or mood in some subgroups of interest. Future studies could specifically recruit a sample of adults with dyskinetic CP to investigate emotion dysregulation and mood disorders amongst adults with dyskinetic CP, given dyskinetic CP frequently involves damage to the basal ganglia [44]. As data for this study were collected by a self-report survey, eligibility criteria were limited to individuals with no/mild intellectual impairment, excluding participation for people with CP who had moderate and severe intellectual impairments. Understanding ER and mood in adults with CP and intellectual impairment is also important and should be considered in future research. There was a high proportion of participants (*n* = 17, 40%) who reported having a diagnosed psychological disorder. This highlights a possible self-selection bias with participants who are aware of, or who are experiencing difficulties in this area of their lives potentially being more likely to respond to the study invitation. Conversely, there may have been an underrepresentation of participants with psychological and emotion regulation difficulties who may have chosen not to participate. Finally, the survey was sent to everyone within the eligible age range on the CP registers. The response rate of 42, although expected for a survey study with no immediate benefit for participants, is low. Results, therefore, may not be representative of the broader CP in adulthood population. Finally, data were collected in 2019 before the COVID-19 pandemic. Marginalised groups, including people with disability, have been disproportionately negatively affected by the COVID-19 pandemic and associated societal and policy changes [45]. Similarly, mental health difficulties have increased as a result of the COVID-19 pandemic; therefore, these rates may now be higher than when the data were collected.

## 5. Conclusions

This is the first study investigating ER amongst adults with CP. Here we observed that the majority of participants had total ER scores within normal limits. However, more participants than expected based on the general population had elevated sub-scale scores, indicating difficulties accepting emotional responses, engaging in goal-directed behaviour, controlling impulses, a lack of emotional awareness, and low belief in and knowledge of strategies to help regulate emotions. Worryingly, a higher proportion of participants reported symptoms of depression than the general population and a relatively high proportion reported symptoms of anxiety. In this study, we found a clear positive relationship between mood and ER scores, whereby participants with elevated depression, anxiety, and stress symptoms reported more ER difficulties than those who did not have elevated depression, anxiety, or stress symptoms. Considering the relationship between mood and ER, use of ER strategies (e.g., mindfulness) may be helpful tools for people with CP and mood difficulties. In the absence of known prevention strategies, screening of emotion regulation and mood difficulties as part of routine health reviews with appropriate onward referral and support should be evaluated.

## Figures and Tables

**Figure 1 jcm-12-02527-f001:**
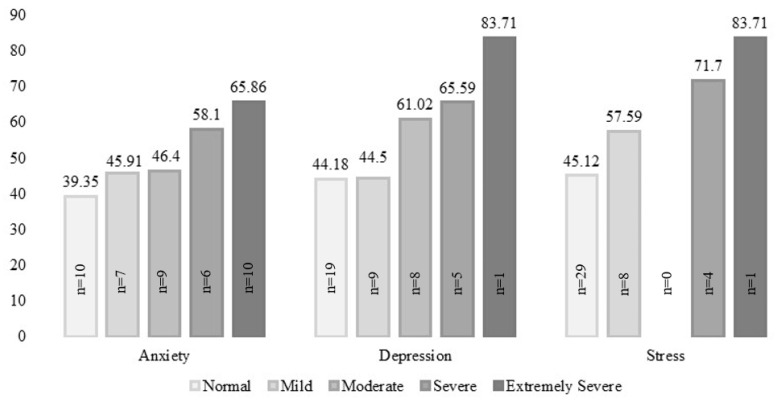
Mean total emotion regulation scores by level of depression, anxiety, and stress.

**Table 1 jcm-12-02527-t001:** Participant characteristics.

Characteristics	Total *n* = 42
Females; *n* (%)	24 (57.1)
Age; Mean (SD); Range	31.50 (13.48); (18–72)
CP Type/topography ^++^ *n* (%)	
Spastic	30 (73.2)
Hemiplegia	15 (50.0)
Diplegia	8 (26.7)
Quadriplegia	7 (23.3)
Dyskinesia	4 (9.7)
Ataxia	1 (2.4)
Mixed	5 (12.2)
Unknown	1 (2.4)
Intellectual Ability *n* (%)	
No impairment	37 (88.1)
Mild impairment	5 (11.9)
Functional Communication Level ^++^ *n* (%)	
Effective communicator	33 (80.5)
Slow but effective communicator	6 (14.6)
Effective communicator with familiar people	2 (4.9)
Inconsistent communicator	0
Seldom effective communicator	0
AAC; N (%)	
Use AAC	6 (14.3)
GMFCS level; *n* (%)	
I	14 (33.3)
II	13 (31)
III	4 (9.5)
IV	6 (14.3)
V	5 (11.9)
Gross Motor Function; *n* (%)	
Ambulant	27 (64.3)
Supported mobility	15 (35.7)
Employment Status ^+^ *n* (%)	
Employed	28 (70.0)
Unemployed	12 (30)
Living Arrangement ^++^ *n* (%)	
Alone	7 (17.1)
Share house	5 (12.2)
With spouse	5 (12.2)
Other family	24 (58.5)
Primary Support Person; *n* (%)	
Parent	22 (52.4)
I do not have a primary support person	13 (31.0)
Partner/spouse	2 (4.8)
Sibling	1 (2.4)
Caseworker/advocate/other	4 (9.6)
Psychological Disorder; *n* (%)	
Diagnosed psychological disorder	17 (40.5)
Routine Services; *n* (%)	
Receiving routine services	18 (42.9)

^+^ N = 40; ^++^ N = 41.

**Table 2 jcm-12-02527-t002:** Means and proportions of depression, anxiety, stress and emotion regulation scores.

**DERS Scores**								
	**Raw DERS**			**T-Scores**				
	**Mean (SD)**			**Mean (SD)**		**WNL ^1^** **N (%)**	**Elevated ^2^** **N (%)**	***p*-Value**
Total	81.93 (30.30)			50.95 (14.87)		38 (90.5)	4 (9.5)	Not able to compute
Nonaccept	14.69 (6.99)			55.52 (14.64)		32 (76.2)	10 (23.8)	<0.001
Goals	13.45 (5.29)			49.67 (12.32)		36 (85.7)	6 (14.3)	0.048
Impulse	12.50 (6.31)			51.28 (13.26)		36 (85.7)	6 (14.3)	0.048
Awareness	13.45 (5.53)			51.52 (13.51)		36 (85.7)	6 (14.3)	0.048
Strategies	17.86 (8.26)			52.25 (12.76)		35 (83.3)	7 (16.7)	0.010
Clarity	9.98 (4.41)			49.04 (11.36)		37 (88.1)	5 (11.9)	0.175
**DASS-21 Scores**								
	**Mean**	**SD**	**Range**	***n* (%) Scoring in Ranges**				
				**Normal**	**Mild**	**Moderate**	**Severe**	**Extremely Severe**
Depression	5.71	3.897	0–18	19 (45.2)	9 (21.4)	8 (19.0)	5 (11.9)	1 (2.4)
Anxiety	7.10	4.143	2–19	10 (23.8)	7 (16.7)	9 (21.4)	6 (14.3)	10 (23.8)
Stress	5.62	4.654	0–20	29 (69.0)	8 (19.0)	0 (0.0)	4 (9.5)	1 (2.4)

^1^ WNL = “within normal limits”; ^2^ Elevated => 1.5 SD from normative mean, 6.68 percent of the normative population would be expected to experience emotion regulation scores 1.5 SD above the mean. Chi-squared goodness-of-fit test performed on DERS scores compared whether sample proportion differed from expected population proportion, based on normative data.

**Table 3 jcm-12-02527-t003:** Non-parametric relationship between depression, anxiety, and stress and total emotion regulation T-scores.

	DASS Low Group			DASS Elevated Group			Change Statistic
	DERS Total T-Score			DERS Total T-Score			
	N	Median	IQR	N	Median	IQR	*p*-Value
Anxiety	17	42.85	16.64	25	56.82	16.86	0.001
Depression	28	43.16	15.94	14	62.18	16.29	<0.001
Stress	37	49.74	17.65	5	71.64	29.27	0.002

Note: ‘Low Group’ refers to scores on the DASS that are ≤mild range; ‘Elevated Group’ refers to scores on the DASS that are ≥moderate range. Mann–Whitney tests were conducted to examine the relationship between Total DERS scores and DASS-21 sub-scales, with significance set to *p* = 0.05.

**Table 4 jcm-12-02527-t004:** Non-parametric examination of depression, anxiety, stress, and emotion regulation difficulties by socio-demographic factors.

	DERs Total			Depression			Anxiety			Stress		
	Median	IQR	*p*-Value	Median	IQR	*p*-Value	Median	IQR	*p*-Value	Median	IQR	*p*-Value
Gross Motor Function												
Ambulant	52.90	10.65	0.232	6.00	5.00	0.225	7.00	5.00	0.117	4.00	5.00	0.518
Supported mobility	43.46	19.08		4.00	3.00		5.00	5.00		4.00	6.00	
Gender												
Male	46.30	26.94	0.416	6.50	6.25	0.289	7.00	7.50	0.574	5.50	7.00	0.898
Female	52.81	19.91		4.00	3.00		6.50	3.75		4.00	3.75	
CFCS												
Effective communicator	49.97	20.75	0.224	5.00	6.25	0.393	6.00	6.00	0.368	4.00	5.25	0.519
Reduced communication	54.36	24.26		6.50	5.75		8.00	8.25		7.00	6.25	
Employment Status ^+^												
Employed	52.81	21.18	0.859	5.00	6.75	0.711	6.50	4.75	0.859	4.00	4.75	0.801
Unemployed	48.26	18.81		5.00	4.75		7.00	6.50		5.00	6.50	
Diagnosed psychologicaldisorder												
yes	54.92	9.93	0.018	6.00	5.00	0.091	7.00	6.00	0.033	6.00	4.50	0.008
no	42.85	21.67		4.00	6.50		5.00	5.50		3.00	6.00	
Receiving routine services												
yes	51.46	21.84	0.722	4.50	5.50	0.498	6.00	6.25	0.601	4.00	4.75	0.344
no	51.97	20.84		5.00	5.75		7.00	5.25		6.00	5.00	

Note: Elevated DASS scores are scores ≥ Moderate range; Elevated DERs total scores are T-scores ≥ 1.5 SD above the mean; ^+^ N = 40; Mann–Whitney nonparametric tests performed.

## Data Availability

The data presented in this study are available on request from the corresponding author. The data are not publicly available due to ethical restrictions.

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
