# Peer review of "Emotion Regulation Is Associated with Anxiety, Depression and Stress in Adults with Cerebral Palsy"

_jcm, 2023, doi:10.3390/jcm12072527_

Round 1

Reviewer 1 Report (Previous Reviewer 2)

Thank you for the responses and explanation for no changes, which make perfect sense. The manuscript looks great! 

Author Response

Thank you, we really appreciate your time.

Reviewer 2 Report (New Reviewer)

Thank you for the opportunity to review this manuscript. The manuscript is strong and adds valuable information regarding emotional regulation among those with CP. Below are some minor comments:

- Missing references for lines 75 to 77 

- Editorial mistakes throughout (e.g., data is plural, than vs. then) 

- Implication for practice can be expanded

Author Response

  • Missing references for lines 75 to 77 

Thank you, we have added appropriate references.

  • Editorial mistakes throughout (e.g., data is plural, than vs. then) 

Thank you. We have made these changes throughout.

  • Implication for practice can be expanded

Thank you for this suggestion. Whilst we would like to expand on implications for practice, another reviewer noted their concerns that our results may not be generalisable and the study did not include an intervention, recommending caution with any recommendations for practice. Hence, we have limited our recommendations to suggestions for further research throughout and some more general suggestions around implications for practice (see discussion).

Reviewer 3 Report (New Reviewer)

Overall, this is an interesting paper which adds to our gap in knowledge about the psychiatric comorbidities in CP.

The authors spend almost half of the introduction on a discussion of neuroanatomical correlates of ER.  As it stands, I was expecting the paper to try to correlate the presence of ER with imaging or CP subtypes. As this purpose of this paper was to obtain preliminary data, I suspect it was not powered to look at this question.  I would make sure the reasons for not exploring that question are explicitly stated somewhere. It is briefly alluded to in line 320-21, but I almost missed it, and instead the repeated emphasis on BG injury and ER seemed out of place. 

I think it is reasonable to nod at neuroanatomical concepts in the introduction, but I would recommend primarily focusing on this in the discussion section.  I appreciated the authors’ theory that Dyskinetic CP would be disproportionately associated with ER as compared to other forms of CP.  All that said, we know that injury to cortico-thalamic tracts is a  better predictor of CP than is injury to motor tracts – there may be similar surprises as we get closer to localization neuropsychiatric conditions in CP. 

The authors mention TBI and Parkinson’s disease as models linking ER to basal ganglia disease; however, these are disorders generally acquired later in life and may not be as relevant to a developmental disorder like CP.  Examples of other developmental disorders that connect ER with BG disfunction (such as Tourette syndrome) may be worth considering.

While the paper focuses on ER and its association with depression and anxiety, ER is also associated with executive dysfunction and ADHD. ADHD is a common comorbidity in CP.  While ADHD is not within the scope of this study, its common association with CP as well as its co-occurrence with difficulty in emotional learning and later psychiatric disorders, including anxiety, depression, and ER should be acknowledged. 

One of the most interesting things this paper adds is that, overall, the stress level in the included subjects was not higher than figures estimated in the general population.  I would consider this more in the discussion, as this is an unexpected finding.  Potential limitations which could explain this include the fact that the study population was a group of adult patients in a CP registry.  That alone selects for people who can make it to a clinic, and thus for people with higher resources (many adults with CP get lost).  Furthermore, it is possible that patients with higher resources, and thus less stressors, are more likely to respond to a survey.  I would be curious to see what stress, ER, anxiety, and depression scales looked like in the subpopulation that was not captured.  The authors mention that there may be selection bias, but I think it is important to emphasize this as it is particularly important in interpretation of results and may mean the authors are only hitting the tip of the iceberg of this problem.

I’m curious as to why the authors feel that non-acceptance of emotional response was so high.  I have seen this in my own patients frequently and have my own theories and would be curious as to what theirs are.

The authors mentioned several potential contributors to ER in their discussion.  One factor not considered is emotional learning.  As alluded to above, CP is defined by its onset in infancy, and thus we cannot think of it in the same way we think about adult-onset conditions.  The presence of disability and brain injury on childhood hijacks development in complex ways, including the natural trajectory of emotional intelligence.  I would have liked to see this considered.

Finally, do not forget the oxford comma.

Author Response

  • Overall, this is an interesting paper which adds to our gap in knowledge about the psychiatric comorbidities in CP.

Thank you.

  • The authors spend almost half of the introduction on a discussion of neuroanatomical correlates of ER.  As it stands, I was expecting the paper to try to correlate the presence of ER with imaging or CP subtypes. As this purpose of this paper was to obtain preliminary data, I suspect it was not powered to look at this question.  I would make sure the reasons for not exploring that question are explicitly stated somewhere. It is briefly alluded to in line 320-21, but I almost missed it, and instead the repeated emphasis on BG injury and ER seemed out of place. 

Thank you, we have removed a little of this text in the background section and have noted this issue as a limitation in the strengths and limitations section.

  • I think it is reasonable to nod at neuroanatomical concepts in the introduction, but I would recommend primarily focusing on this in the discussion section.  I appreciated the authors’ theory that Dyskinetic CP would be disproportionately associated with ER as compared to other forms of CP.  All that said, we know that injury to cortico-thalamic tracts is a better predictor of CP than is injury to motor tracts – there may be similar surprises as we get closer to localization neuropsychiatric conditions in CP. 

Thank you, we have removed some of the neuroanatomical introduction and increased the focus on CP and co-occurring conditions as suggested below.

  • The authors mention TBI and Parkinson’s disease as models linking ER to basal ganglia disease; however, these are disorders generally acquired later in life and may not be as relevant to a developmental disorder like CP.  Examples of other developmental disorders that connect ER with BG disfunction (such as Tourette syndrome) may be worth considering.

Thank you, we have added this to the introduction.

  • While the paper focuses on ER and its association with depression and anxiety, ER is also associated with executive dysfunction and ADHD. ADHD is a common comorbidity in CP.  While ADHD is not within the scope of this study, its common association with CP as well as its co-occurrence with difficulty in emotional learning and later psychiatric disorders, including anxiety, depression, and ER should be acknowledged. 

Thank you, we have added this to the introduction.

  • One of the most interesting things this paper adds is that, overall, the stress level in the included subjects was not higher than figures estimated in the general population.  I would consider this more in the discussion, as this is an unexpected finding.  Potential limitations which could explain this include the fact that the study population was a group of adult patients in a CP registry.  That alone selects for people who can make it to a clinic, and thus for people with higher resources (many adults with CP get lost).  Furthermore, it is possible that patients with higher resources, and thus less stressors, are more likely to respond to a survey.  I would be curious to see what stress, ER, anxiety, and depression scales looked like in the subpopulation that was not captured.  The authors mention that there may be selection bias, but I think it is important to emphasize this as it is particularly important in interpretation of results and may mean the authors are only hitting the tip of the iceberg of this problem.

Thank you for highlighting this issue. The CP registers across the three states recruit through public hospitals, community disability services and also allow self- registration by families and people with lived experience of CP. We agree that it is possible that participants are more likely to be those with higher resources or capacity to respond. We are unsure how we could capture the responses of people who did not wish to participate in this study. We agree that the self-selection bias could have led to over or under representation. We have added additional text to this effect in the strengths and limitations section.

  • I’m curious as to why the authors feel that non-acceptance of emotional response was so high.  I have seen this in my own patients frequently and have my own theories and would be curious as to what theirs are.

Thank you raising this discussion point. We are also curious about this. Looking at the limited literature we feel it may be associated with the high frequency of psychological symptoms that the sample reported, given that non acceptance of emotional responses is associated with psychopathology. Non acceptance of emotional responses hampers application of strategies that might be protective for mental health. We wonder if perhaps there would be cross loading of items between a non-acceptance ER factor and mood factor, which could explain the elevated results of non-acceptance in this sample. There is little about this that we can find in the literature, and we feel it inappropriate to comment on this in more detail within the manuscript. We would be interested to hear your thoughts.

  • The authors mentioned several potential contributors to ER in their discussion.  One factor not considered is emotional learning.  As alluded to above, CP is defined by its onset in infancy, and thus we cannot think of it in the same way we think about adult-onset conditions.  The presence of disability and brain injury on childhood hijacks development in complex ways, including the natural trajectory of emotional intelligence.  I would have liked to see this considered.

Thank you. We agree and had considered this concept. It was introduced in lines 80-82 of the introduction and have now added it to the discussion.

  • Finally, do not forget the oxford comma.

Thank you. We have made this change throughout.

This manuscript is a resubmission of an earlier submission. The following is a list of the peer review reports and author responses from that submission.

Round 1

Reviewer 1 Report

This is great topic, the paper is well written, tools excellent.

There are several issues.

1) title is not helpful--should be something like  "Emotional dysregulation increases with increased stress, anxiety, and depression in adults with cp"

2) sampling. There is no comparison group (adults with similar demographics and no CP ) and no stratified or targeted sampling so the results are biased and not generalizable.  This is a huge issue in interpretation of the information. 

I do value the approach of collecting the data and breaking the subjects into sub groups for analysis on specific outcome measures. However, in this instance, I believe the sample size is too small to do any comparisons of socio demographics. There is no power analysis no sample size justification for the analyses 

The value in this study was that the data was collected on a group of adults with CP and as expected it was shown anxiety, depression, and stress increase with decreased Emotional regulation.

The discussion makes overreaching claims. There are population based studies that show mood is an issue in adults with CP.  

this study primarily adds to our knowledge that in a group of people with CP ER decreased with mood issues.  

Reviewer 2 Report

This is an excellent study that generates novel information relating to emotion regulation in adults with CP, which likely plays a role in many aspects of health and function and thus an important concept to study. The authors do a great job harnessing the data that balances generating helpful information with the small sample size and self-report e-surveys. The statistics, analytic approach, and variable decisions (e.g., grouping) seem appropriately aligned with the goals of the study and the available data. The authors also do a great job with describing notable limitations that should be weighed when interpreting the results, which allows for a focused interpretation. They also appropriately refer to their data as preliminary and do not appear to over-interpret such results. I have 1 major comment and a handful of minor comments, all of which can be easily remedied with a simple revision or rebuttal. I hope my comments can improve an already well-balanced, informative study.

Major

If the authors are within the allowable number of figures/tables, please consider bringing Supplemental Table 1 as Table 4. This information is part of the objectives and should be presented as part of the main text if possible. If the number of figures/table limit is reached currently, then presenting this in the supplement makes sense. Alternatively, since Table 3 and Figure 1 present similar information, Table 3 can probably be put to the supplement with the findings simply described in the text along with the p-values for differences. This would allow pulling in Supplemental Table 1, which is a nice table and important to understand in this study.

Minor

Abstract: consider referencing somewhere that higher scores reflect greater difficulties in emotion regulation. There are lots of scales and higher values do not always mean more of the construct being measured. One example to help interpret could be something like, line 23: “…proportion of participants experienced elevated scores (i.e., more difficulties with emotion regulation) that would be…”

I really appreciate the mention of construct validity and reliability statistics for the DASS-21.

The authors mention that validity and reliability of the DERS is well established. Can the authors provide a solid reference or two? Better yet, can the authors provide a reliability/validity statistic in parentheses like they did for the DASS-21?

Lines 199-200: 2x2 tables in SPSS can be compared if a cell has n<5 using Fisher’s Exact test. I am a SAS user. Otherwise, I would provide the steps to conduct this test, but I am sure it is easy to find online.

Table 1: does the asterisk in the “Elevated” column indicate statistical significance at P<0.05? I don’t see that explained in the table’s footnote. If you have the p-values, the asterisk indicating statistical significance seems redundant.  

Figure 1 is an excellent presentation of the results.

The responses are all self-report and the authors appropriately justify this approach and note it in the limitations. One set of measures that I wonder about is how accurate the collection of GMFCS, CFCS, and AAC are. In the U.S., many adults with CP are unfamiliar with GMFCS, even those that work in the research field. Perhaps the use of these scales is different in other countries. To enhance efficiency, I will provide 2 responses relating to this. FIRST, if GMFCS, CFCS, and AAC are not well-known and difficult to self-report, then one issue would be the accuracy of obtaining these values. This would not impact the main analyses, which is why I consider this a “minor” comment, as it would only impact a subset of analyses within the secondary analysis that compares the outcomes across ambulatory vs. non-ambulatory groups (which were derived from the GMFCS), CFCS, etc. Not much one can do here, except note it in the Limitations. Perhaps at line 313, this can be drawn out more as this limitation seems to fall within the broader “self-report” limitation. SECOND, if GMFCS, CFCS, and AAC are well-known and thought to be reliable from self-report, then perhaps a statement in the Methods section that helps international readers understand this.